# Environmental Stress Symptoms during Heat Acclimatization, Heat Acclimation, and Intermittent Heat Training

**DOI:** 10.3390/ijerph20043219

**Published:** 2023-02-12

**Authors:** Ciara N. Manning, Courteney L. Benjamin, Yasuki Sekiguchi, Cody R. Butler, Michael R. Szymanski, Rebecca L. Stearns, Lawrence E. Armstrong, Elaine C. Lee, Douglas J. Casa

**Affiliations:** 1Korey Stringer Institute, Department of Kinesiology, University of Connecticut, Storrs, CT 06269, USA; 2Department of Kinesiology, Samford University, Birmingham, AL 35226, USA; 3Sports Performance Lab, Department of Kinesiology and Sport Management, Texas Tech University, Lubbock, TX 79430, USA; 4Special Warfare Human Performance Squadron, Lackland Air Force Base, San Antonio, TX 78236, USA; 5Human Performance Lab, Department of Kinesiology, University of Connecticut, Storrs, CT 06269, USA

**Keywords:** heat stress, heat adaptation, symptoms, thermal sensation, perceptual adaptation

## Abstract

Background: Athletes training in heat experience physiological and perceptual symptoms that risk their safety and performance without adaptation. Purpose: We examined the changes in environmental symptoms, assessed with the Environmental Symptoms Questionnaire (ESQ), during heat acclimatization (HAz), heat acclimation (HA), and intermittent heat training (HT). Methods: Twenty-seven participants (mean ± standard deviation [M ± SD], age of 35 ± 12 y, VO_2max_ of 57.7 ± 6.8 mL·kg^−1^·min^−1^) completed five trials involving 60 mins of running (60% vVO_2max_) followed by a 4 km time trial in heat (M ± SD, temperature of 35.5 ± 0.7 °C, humidity of 46.4 ± 1.5%). The trials occurred at baseline, post-HAz, post-HA, at week 4 of HT (post-HT4), and at week 8 of HT (post-HT8). The participants completed HT once/week (HT_MIN_), completed HT twice/week (HT_MAX_), or did not complete HT (HT_CON_). ESQ symptoms, thermal sensation (TS), and heart rate (HR) were measured pre- and post-trial. Results: Post-ESQ symptoms improved post-HA (3[0.40, 4.72], *p =* 0.02) and post-HAz (3[0.35, 5.05], *p =* 0.03) from baseline. During HT, symptoms improved in the HT_MAX_ group and worsened in the HT_MIN_ and HT_CON_ groups. Symptoms improved in the HT_MAX_ group versus the HT_CON_ group at post-HT8 (4[1.02, 7.23], *p =* 0.012). Higher TS and HR values were weakly associated with ESQ symptoms during HT (*r =* 0.20, *p =* 0.04), only explaining 20% of variance. Conclusions: ESQ symptoms improved during HAz, HA, and HT 2x/week. ESQ symptoms were not statistically correlated with HR during exercise heat stress. TS was not sensitive to detecting adaptation and did not subjectively change. The ESQ may be valuable in monitoring adaptation and may contribute to performance post-acclimation.

## 1. Introduction

Environmental conditions, such as high heat and humidity, place a considerable amount of strain on athletes, requiring them to adapt to various circumstances to maintain their safety and to achieve optimal performance [1]. The previous literature has examined the use of an Environmental Symptoms Questionnaire (ESQ) scale to measure the incidence and severity of the symptoms (the term “symptoms” encompasses physiological symptoms, moods, performance, and cognitive capability) that athletes may experience when exercising in extreme environmental conditions [2]. The questionnaire encompasses symptoms such as “I feel hot”, “I feel lightheaded”, “I feel nauseous”, “I feel ‘goose bumps’ or chills”, and “I can play at my best” [1,2,3,4]. Athletes report the level at which they are experiencing the aforementioned and other heat-stress-related symptoms on a scale from zero (“not at all”) to five (“extreme”). The thermal sensation scale (TS) is another scale commonly used along with the ESQ in which athletes indicate how hot or cold they feel from 0.0 (“unbearably cold”) to 8.0 (“unbearably hot”), with 4.0 indicating “comfortable” [5,6]. TS helps to quantify an athlete’s perception of the surrounding environment [6]. Together, the ESQ and TS work to quantify the subjective ratings of thermal strain and the subjective experiences of physiological symptoms [6].

Researchers have found that the symptoms of heat strain, measured using the ESQ and TS, are exacerbated significantly following exercise in the heat [6,7,8,9,10,11]. Yeargin et al. reported that symptoms including “I have a headache”, “I feel dizzy”, “I feel nauseous”, and “I feel hot” were consistently reported in youth athletes during practices as well as during games in the heat [10]. Along with these symptoms, “thirst”, “tiredness”, and “trouble concentrating” were also frequently reported [10]. TS has been determined to be the biggest indicator of skin temperature and of the rate of skin temperature change and has been found to correlate with other physiological variables, such as heart rate (HR) and rectal temperature (Trec) [1,3,5,12,13]. Therefore, assessing environmental stress symptoms among athletes is important, as this may influence their performance or be indicative of the onset of exertional-heat-related illnesses [7,14,15].

Athletes can best prepare their bodies and enhance their ability to adapt to extreme environmental conditions, such as high ambient temperature, through processes such as heat acclimatization (HAz) and heat acclimation (HA) [16,17]. HAz involves gradual exposure outdoors to the heat and humidity of the natural environment over a period of time [16,17]. HA involves gradual exposure to heat and humidity indoors in a controlled laboratory environment for a period of time [16,17]. These two methods differ in both their setting and their level of control of environmental conditions, and the level of heat and humidity may vary between exposures outdoors; however, when conducted indoors, the conditions can be adjusted to remain at a consistent temperature and humidity. Many physiological adaptations occur due to HAz and HA, including improvements in aerobic fitness, decreases in HR at a given work intensity, decreases in the rate of the rise in core body temperature and lower core body temperature at a given intensity, decreases in perceived exertion at a given intensity, and increases in sweat rate [16,17,18,19]. HAz and HA have also been found to improve individuals’ perception of thermal strain [17]. Researchers have previously explored ESQ adaptations following HAz or HA [1,3,20,21].

Through previous research, it has been found that ESQ symptoms are exacerbated during the initial days of an HA protocol and are improved during the following days [1,3,20]. However, other researchers have reported no changes in ESQ symptoms following an HAz or HA protocol [21]. Additionally, as part of a larger study, Vanscoy et al. investigated ESQ symptoms before and after heat stress tests over the course of a 10-day HA induction period followed by either intermittent heat exposure or no heat exposure [3]. This study also examined the HA decay period for the twenty-five days thereafter with no heat exposure [3]. These researchers found that ESQ symptoms improved following 10 days of HA and that those completing intermittent heat exposure over the following 4 days were more able to sustain symptomatic and physiological adaptations [3]. It was reported that the changes in TS over the course of this HA protocol and the 25-day decay period thereafter were largely accounted for by the changes in ESQ symptoms [3].

No previous research has investigated the effects of intermittent heat training (HT) or of combined acclimatization and acclimation on ESQ symptoms. Therefore, the primary purpose of this study was to investigate the changes in ESQ symptoms over the course of Haz, HA, and HT. We hypothesized that, if an individual’s perceptions, as measured by the ESQ, accurately reflected the progression of their physiological adaptations independent of breaks in training and of the nature of environmental exposure (i.e., laboratory vs. natural), we would observe a higher ESQ score post-trial versus pre-trial regardless of the intervention and that post-trial symptoms would incrementally decrease over the course of HAz, HA, and HT, respectively. A secondary aim of this study was to assess the relationships between ESQ symptoms and TS, HR, and T_rec_ and to investigate if TS alone or if TS, HR, and T_rec_ combined were predictive of post-trial ESQ symptoms. We hypothesized that, if TS, HR, and T_rec_ were strongly correlated with ESQ symptoms, future research into using the ESQ as a noninvasive indicator of physiological responses could be justified. 

## 2. Materials and Methods

### 2.1. Ethical Approval

Procedures in this study were approved by the <<removed for review>> Institutional Review Board. Participants provided both written and informed consent and were medically cleared prior to participation. This study took place in <<removed for review>>. Data presented within this manuscript are part of a larger study that focused on physiological and performance measures relative to this heat training protocol; however, the current study investigated different hypotheses and data focusing on environmental stress symptoms reported in the ESQ before and after heat stress trials during HAz, HA, and HT [22,23,24,25,26].

### 2.2. Participants

Twenty-seven aerobically fit males (age of 35 ± 12 y, body mass of 72.6 ± 8.8 kg, VO_2max_ of 57.7 ± 6.8 mL·kg^−1^·min^−1^) completed five heat stress trials involving 60 min of treadmill running (approximately 60% vVO_2max_) followed by a 4 km time trial on a treadmill (T150; COSMED, Traunstein, Germany) in the heat (ambient temperature of 35.5 ± 0.7 °C, relative humidity of 46.4 ± 1.5%, wet bulb globe temperature (WBGT) of 29.4 ± 0.4 °C, wind speed of 4.0 ± 0.1 mph). The baseline trial was performed before participants received any heat exposure in the laboratory or in an outside environment. Before the trial, participants provided a urine sample to ensure a state of euhydration via urine-specific gravity (USG) of < 1.025. Before and after each heat stress trial, participants were given a modified ESQ (ESQ-14) to assess the extent to which they experienced environmental stress symptoms, such as “I feel hot”, “I feel lightheaded”, “I feel nauseous”, “I feel ‘goose bumps’ or chills”, and “I can play at my best”, on a scale of 1 (“not at all”) to 6 (“extreme”). ESQ scores refer to the individual values indicated by the athletes for each symptom, and total ESQ score refers to the calculated total sum of values for each athlete at the respective timepoints. During heat stress trials, T_rec_ was measured using MP160 (BIOPAC Systems Inc., Goleta, CA, USA) and HR using H10^®^ (Polar Electro™, Kempele, Finland). Additionally, TS was measured on a 0.0–8.0 Likert scale (0.0 being unbelievably cold, 8.0 being unbelievably hot) throughout the trial. 

Participants completed a baseline trial at the start of the study and then trained autonomously during the summer with no specific instruction regarding duration, intensity, or modality (Haz of 109 ± 9 days). During HAz, participants recorded each training session using their preferred wearable technology. Several variables, including total distance covered, training time, and average HR, and environmental conditions, including ambient temperature, relative humidity, and WBGT, were recorded. After HAz, participants performed a heat stress test to examine adaptations that resulted from HAz (post-HAz). Then, participants completed 5-day HA induction following HAz. The HA sessions involved participants exercising at a hyperthermic internal body temperature (between 38.50 °C and 39.75 °C) for 60 min. To do this, participants began exercising at a higher intensity (~ 70% vVO_2max_) to increase T_rec_ rapidly to 38.5 °C and then adjusted the intensity to maintain a state of hyperthermia for 60 min. This type of HA method is defined as “hyperthermic zone HA” (HZHA). Participants performed another trial after HA induction to investigate the adaptations that resulted from this HA (post-HA). Including both HAz and HA in this protocol was important for participants to acquire natural heat exposure outdoors as well as a controlled and consistent amount of heat exposure indoors to elicit the greatest adaptations possible before beginning HT in an attempt to maintain these adaptations.

Post-HA, participants performed 8 weeks of HT. Participants were randomly assigned to 3 groups during HT, completing HT 2x per week (HT_MAX_) for a total of 16 sessions, completing HT 1x per week (HT_MIN_) for a total of 8 sessions, or completing no HT sessions (HT_CON_). More data on physiological variables of groups in this study were reported in other manuscripts. The heat stress protocol used for HT was the same as that used for the HA sessions. Heat stress frequencies (twice or once per week) were selected to explore minimal amounts of heat stress needed following acclimation to maintain adaptations. Participants completed trials at week 4 of HT (post-HT4) and at week 8 of HT (post-HT8) to investigate the adaptations during HT. See Figure 1 below for experimental design. 

This figure indicates the timeline and description of different phases of this study along with measurements that were collected during each of the heat stress trials. All measurements listed for heat stress trial 1 were collected during every heat stress trial.

### 2.3. Statistical Analyses

All data were assessed for normality and sphericity prior to analyses using SPSS (IBM version 26.0). Repeated measures ANOVAs assessed changes in pre-trial, post-trial, and post-pre-trial ESQ scores over the course of HAz, HA, and HT. Data from ANOVAs were reported as *F*-values and *p*-values. Greenhouse–Geisser corrections were used when the assumption of sphericity was violated. Post hoc independent and dependent *t*-tests assessed between-group and within-group differences. Cohen’s d (equal sample size) and Hedges’ g (unequal sample size) were used to determine the magnitude of differences: small (0.2–0.49), medium (0.5–0.79), or large (> 0.8) effects [27]. Data were reported as mean differences (95% confidence interval), *p*-values, and effect sizes (MDs [95%CI], *p*-values, ESs). Spearman’s bivariate correlations assessed whether relationships existed between post-trial ESQ symptoms, post-trial HR, post-trial TS, and post-trial Trec. Correlation coefficient thresholds were used at 0.1, 0.3, 0.5, 0.7, and 0.9, indicating small, moderate, large, very large, and extremely large associations, respectively, along with the use of *p*-values to indicate statistical significance [28]. Data from correlations were reported as *r*-values and *p*-values. Linear and stepwise regressions were used to investigate if TS alone or if TS, HR, and T_rec_ combined were predictive of post-trial ESQ symptoms. Data from regressions were reported as *r*-values and *p*-values. Significance was set at *p* ≤ 0.05 a priori. 

## 3. Results

### 3.1. HAz and HA 

As previously mentioned, these data were part of a larger study that specifically investigated the physiological effects occurring in HR, Trec, and sweat rate (SR) as well as the performance measures, and more detailed analyses can be found in [22,23,24,25,26]. The average duration of exercise during each HAz session was 56.38 ± 72.66 min for the running sessions and 91.67 ± 69.27 min for the cycling sessions. During these sessions, the average running HR was 140 ± 15 bpm, and the average cycling HR was 128 ± 16 bpm. The average WBGT recorded during HAz was 22.31 ± 4.23 °C for the running sessions and 23.68 ± 3.96 °C for the cycling sessions. The HA sessions involved an average duration of exercise of 82 ± 5 min. T_rec_ averaged 38.83 ± 0.25 °C for the entire HA session and 39.17 ± 0.17 °C for the 60 min hyperthermic period of the HA session. During these sessions, the average HR was 132 ± 11 bpm for the entire session and 132 ± 12 bpm for the 60 min hyperthermic period. 

Decreases in HR and T_rec_ and increases in SR following HAz and HA confirmed the adaptations. The average HR and T_rec_ post-HA (HR of 134 ± 11 bpm, T_rec_ of 38.03 ± 0.39 °C) were significantly lower compared to both of them at baseline (HR of 143 ± 12 bpm, *p* < 0.001; T_rec_ of 38.29 ± 0.37 °C, *p* = 0.005) and post-HAz (HR of 138 ± 14 bpm, *p* = 0.013; T_rec_ of 38.25 ± 0.42 °C, *p* = 0.009). The average HR post-HAz (*p* = 0.002) was significantly lower compared to baseline. The sweat rate was significantly higher post-HA (1.93 ± 0.47 L·h^−1^) compared to post-HAz (1.76 ± 0.43 L·h^−1^, *p* = 0.027). 

### 3.2. ESQ Symptoms 

No significant differences in pre-ESQ symptoms occurred during HAz, HA, or HT (*p* > 0.05). Post-ESQ symptoms improved significantly from baseline to post-HA (3[0.40, 4.72], *p =* 0.02, ES = 0.66) and from post-HAz to post-HA (3[0.35, 5.05], *p =* 0.03, ES = 0.53), but not from baseline to post-HAz (*p* > 0.05). Post-ESQ symptoms changed significantly during HT (*F =* 3.78, *p =* 0.048) (Table 1). Specifically, from post-HA to post-HT8, ESQ symptoms changed significantly independent of the groups (1[−6.91, −0.34], *p =* 0.04, ES = 1.31). Symptoms were significantly lower in the HT_MAX_ group compared to the HT_CON_ group (4[1.02, 7.23], *p =* 0.012, ES = 1.25) at post-HT8. During HT, symptoms improved in the HT_MAX_ group and worsened in the HT_MIN_ and HT_CON_ groups. Statistical significance was only found in the HT_CON_ group with significant increases from post-HA to post-HT8 (SEM = 1.39 [−6.91, −0.34], *p* = 0.04) (Table 2). 

The symptoms that were rated with the highest scores consistently among all three trials following each stage of heat exposure (HAz, HA, and HT) were “I feel thirsty”, “I feel tired”, and “I feel hot”. No statistically significant differences were found between individual symptoms (*p* > 0.05) (Figure 2a). 

Changes in ESQ symptoms were calculated by subtracting the pre-trial values from the post-trial values. Changes in ESQ symptoms improved significantly throughout induction (HAz, HA) (*F =* 6.88, *p =* 0.002) and HT (*F =* 3.80, *p =* 0.04). Post-pre-trial symptoms worsened from baseline to post-HAz (3[0.83, 5.39], *p =* 0.01) and improved from post-HAz to post-HA (3[1.22, 5.38], *p <* 0.001). During HT, post-pre-trial symptoms improved significantly from post-HA to post-HT4 (−3[−6.11, −0.24], *p*= 0.04) and from post-HA to post-HT8 (−2[−4.26, −0.62], *p=* 0.01) independent of the groups. Post-pre-trial symptoms were significantly higher at post-HT8 in the HT_MIN_ group versus the HT_MAX_ group (5[0.25, 9.66], *p=* 0.04). Post-pre-trial symptoms worsened significantly in the HT_CON_ group from post-HA to post-HT8 (1[−6.94, −2.31], *p <* 0.001) (Figure 2b). 

### 3.3. ESQ Symptoms, TS, HR, and T_rec_

Linear regressions revealed that, post-HA, higher TS predicted more severe ESQ symptom scores (*r* = 0.14, *p <* 0.001 [95%CI, 1.41, 5.01]). Specifically, 13.8% of the variance in post-trial ESQ symptoms could be explained by TS. When looking at multiple variables, such as TS, HR, and T_rec_, a stepwise regression indicated that higher TS and higher HR values predicted higher ESQ symptoms, explaining 20.8% of variance (*r =* 0.21, *p =* 0.01) (Figure 3). Interestingly, T_rec_ was not found to predict ESQ symptoms (*p >* 0.05). During HT, linear regressions revealed that higher TS predicted more severe ESQ symptoms independent of the groups (*r =* 0.07, *p =* 0.01 [95%CI, 0.59, 4.42]). When combined, a stepwise regression indicated that higher TS and higher HR values predicted more severe ESQ symptoms during HT (*r =* 0.20, *p =* 0.04) (Figure 4). During HT, T_rec_ was also not predictive of ESQ symptoms (*p >* 0.05). 

A moderate correlation was found between post-trial ESQ symptoms and post-trial HR at baseline (*r =* 0.39, *p =* 0.04), post-HA (*r =* 0.38, *p =* 0.05), as well as at post-HT8 independent of the groups (*r =* 0.49, *p =* 0.01). An extremely large correlation was found between post-trial ESQ symptoms and post-trial HR within the control group post-HA (*r =* 0.90, *p =* 0.00) and at post-HT8 (*r =* 0.70, *p =* 0.05). No correlations with HR and ESQ symptoms were found within the other groups (*p* > 0.05). Post-trial T_rec_ was not correlated with post-trial ESQ symptoms at any time points (*p* > 0.05). 

## 4. Discussion

The purpose of this study was to examine the changes in environmental stress symptoms (ESQ) pre- and post-trial during HAz, HA, and HT using protocols that incorporate real-life applicable training experiences (intermittently and in both natural and artificial environments). We observed that ESQ symptoms improved significantly over the course of HAz and HA and continued improving when the participants participated in HT twice per week. HT once per week or no HT following HAz and HA was not enough to maintain the ESQ adaptations. We found that the improvements in ESQ symptoms began to return to baseline and even worsen when the participants only participated in HT once per week or not at all following induction. Remarkably, the ESQ is sensitive to the decay of the acclimated/acclimatized state and provides perceptual symptom-related justification for intermittent heat exposure to maintain physiological adaptation. Furthermore, this reflects the sensitivity of the ESQ in determining the optimal dose of heat exposure to facilitate both physiological and perceptual adaptations. This may be important in performance and safety, as it is directly related to how an individual feels and to whether an individual is able to exert their maximal effort during training for optimal benefits or during competitions/missions for maximum performance.

Previous researchers have found that the most consistently reported ESQ symptoms in athletes exercising in the heat were “I have a headache”, “I feel dizzy”, “I feel nauseous”, and “I feel hot” [14]. Symptoms such as “I feel thirsty”, “I feel weak”, and “I have trouble concentrating” were also common [14]. Similarly, in our study, we found “I feel thirsty”, “I feel tired”, and “I feel hot” to be the symptoms reported with the highest values following each trial. Symptoms including “I feel weak” and “I can play at my best” were the next symptoms reported to have the highest values. The qualitative nature of the statistically significant symptoms are important to an individual’s ability to perform with good attention and safety during exercise and are important to providing important insights into the ways that HA, HAz, and HT improve an athlete’s ability to train and compete (e.g., putting forth effort without fatigue, maintaining form and balance without fatigue or dizziness, maintaining a sensitive sense of key urges such as thirst).

Researchers have also previously found correlations between physiological variables, such as HR and T_rec_, and ESQ symptoms [1,3,5,7,8]. We likewise observed a correlation between higher HR and/or TS values and more severe ESQ symptoms. HR and TS explained only ~ 20% of the variance found in ESQ symptoms, and we considered this to be an important aspect of interpretation. ESQ symptoms encompass mood, physical, and cognitive symptoms, and responses to these may be diverse in a subject. Additionally, T_rec_ was not correlated with ESQ symptoms. These results supported our interpretation that the ESQ is not, at this time, a substitute marker for physiological adaptation but is a valuable addition to the unique perceptual experience of each individual. 

Our findings that ESQ symptoms track HA, HAz, and HT periods, including breaks and changes in environmental exposure conditions and types, extend the work of the researchers who have found beneficial adaptations in symptoms in isolated single protocols [1,3,12,17,19,20,21,29]. The ESQ was able to detect the added benefit of HT, and, thus, we interpreted that, in addition to the ESQ providing a unique perception to the holistic (physiological, thermal, and perceptual) adaptation of an individual, the ESQ can also be used to complement efforts to monitor acclimation decay and maintenance with intermittent heat exposure after initial acclimation. 

Our study had a few key limitations directly related to the factors that could influence perceptual symptoms. We aim to study female participants in ongoing and future research. It is possible that other variables that were not controlled for in these analyses, such as sleep and diet, could have impacted ESQ symptoms. However, to minimize the effects of alternative factors, the participants were instructed to practice similar nutritional habits for a period of three days before the trials. Additionally, while the participants were able to utilize multiple modalities of endurance exercise during summer training, including but not limited to running or cycling, HA was limited to treadmill running. Future research could benefit from investigating the responses of the different exercise modalities utilized during HAz and their translation to similar or differing modalities of exercise during HA and HT. Finally, future research should attempt to utilize technology that reports and assesses ESQ symptoms to identify the most feasible and convenient method for the regular monitoring of environmental symptoms during training. 

## 5. Conclusions

In conclusion, improvements in ESQ symptoms occurred following HA and were sustained during HT. Our findings suggested that, when assessing ESQ symptoms, multiple variables should be considered and that monitoring should occur after each practice or training session. It is extremely important that ESQ symptoms are monitored frequently over the course of HAz, HA, and HT, as our study exemplified that these symptoms change over this course and that changes in ESQ symptoms can happen more gradually and sensitively to the stimuli. Practically, this is important for coaches to know, as they can easily monitor athletes’ symptoms using the ESQ to track any changes that may occur as athletes adapt to the heat. In the future, it may be more effective and convenient to have technology that can easily be accessed by athletes, allowing them to report ESQ symptoms following training sessions and practices in the heat. Additionally, it may be beneficial for teams that compete in the heat to continue participating in HT at least twice per week following initial pre-season heat exposure. Continued participation in HT twice per week will help facilitate the maintenance of favorable adaptations in environmental stress symptoms throughout the duration of the season. This will be beneficial in preventing environmental stress symptoms from negatively impacting performance when competing in extreme environments. Monitoring ESQ symptoms daily in addition to physiological measures is pertinent to understanding how athletes are adapting to exercise in the heat during initial and repeated exposure to sustain the optimal levels of all adaptations. 

### Key Points

Heat training twice/week following heat acclimation may be effective in improving and maintaining adaptations in environmental symptoms. More evidence is needed to understand the magnitude of the effect of a specific duration of training on optimizing ESQ adaptations.Environmental symptom changes over the course of heat adaptation happen gradually and sensitively to environmental and exertional stimuli, and these could be a useful and noninvasive monitoring tool to use after every practice, training session, and competition to fully understand the athlete’s status over the course of a season that is not reflected in physiological measures alone.When assessing environmental symptoms and the adaptations of these symptoms, no single physiological variable is fully indicative of these adaptations or is fully capable of reflecting psychological adaptations, and, therefore, a more wholistic approach that encompasses several different variables, beyond physiological variables, should be utilized to fully understand the unique experiences of individual athletes.

## Figures and Tables

**Figure 1 ijerph-20-03219-f001:**
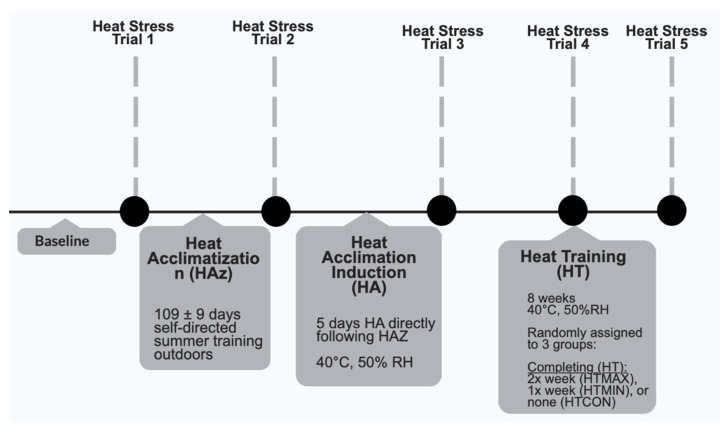
Experimental study design.

**Figure 2 ijerph-20-03219-f002:**
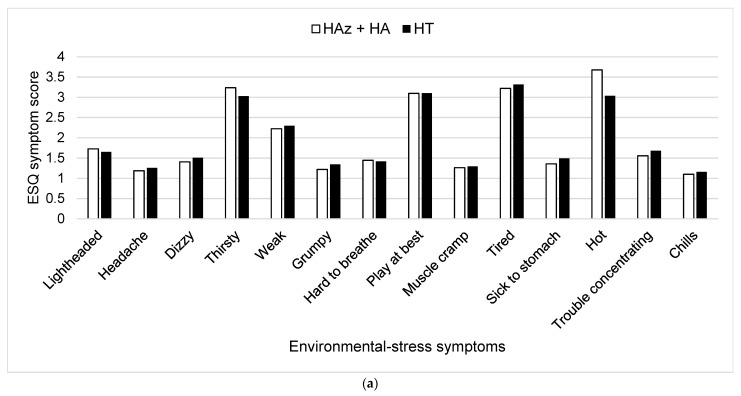
(**a**) Individual post-trial Environmental Symptoms Questionnaire (ESQ) scores following the combination of heat acclimatization (HAz) and heat acclimation (HA) and following heat training (HT). White bars represent average post-trial symptom scores during induction. Black bars represent average post-trial symptom scores during HT. (**b**) Visual representation of change from post-trial to pre-trial Environmental Symptoms Questionnaire (ESQ) symptom scores over the course of heat training (HT) after heat acclimation (post-HA), at week 4 of HT (post-HT4), and at week 8 of HT (post-HT8). Bars represent average scores, and the number value is reported over each bar. Lines represent standard deviations. Brackets and asterisks indicate significant differences in the HT_MIN_ and HT_MAX_ groups at post-HT8 and in the HT_CON_ group from post-HT4 to post-HT8.

**Figure 3 ijerph-20-03219-f003:**
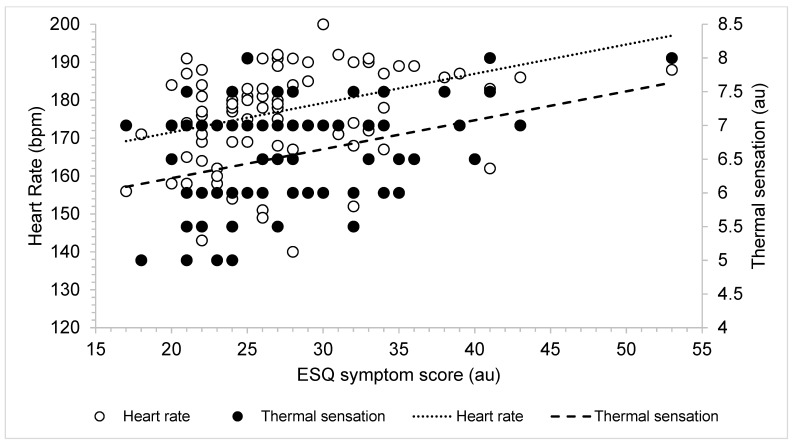
Visual representation of relationship between thermal sensation (TS), heart rate (HR), and Environmental Symptoms Questionnaire (ESQ) symptoms during heat acclimatization and heat acclimation. The dotted line indicates line of best fit for HR, and the dashed line indicates the line of best fit for TS.

**Figure 4 ijerph-20-03219-f004:**
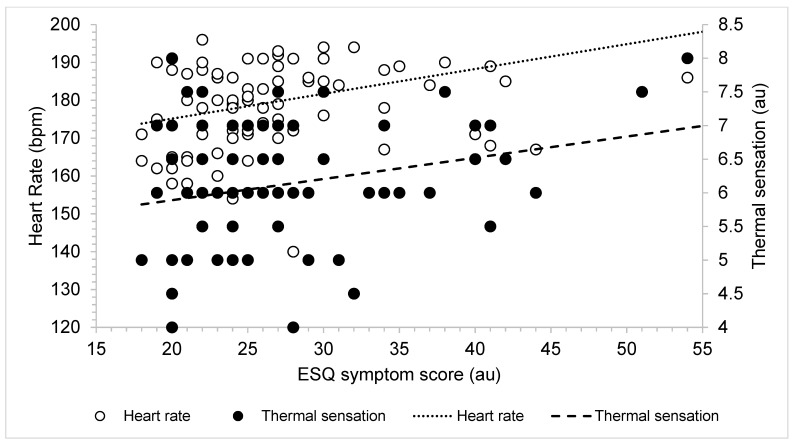
Visual representation of trend in thermal sensation (TS), heart rate (HR), and Environmental Symptoms Questionnaire (ESQ) symptoms over the course of heat training. The dotted line indicates line of best fit for HR, and the dashed line indicates the line of best fit for TS.

**Table 1 ijerph-20-03219-t001:** Means and standard deviations (Ms ± SDs) of post-trial Environmental Symptoms Questionnaire (ESQ) scores at baseline, after heat acclimatization (post-HAz), and after heat acclimation (post-HA). Scores for each item can range from a value of 1 (“not at all”) to 6 (“extreme”).

	Baseline	Post-HAz ^a^	Post-HA ^b^
**Lightheaded**	2 ± 1	2 ± 1	1 ± 1
**Headache**	1 ± 1	1 ± 1	1 ± 0
**Dizzy**	1 ± 1	2 ± 1	1 ± 1
**Thirsty**	4 ± 1	3 ± 1	3 ± 1
**Weak**	2 ± 1	2 ± 1	2 ± 1
**Grumpy**	1 ± 0	1 ± 1	1 ± 1
**Hard to breathe**	2 ± 1	1 ± 1	1 ± 0
**Play at my best**	3 ± 1	3 ± 1	4 ± 1
**Muscle cramp**	1 ± 0	1 ± 1	1 ± 0
**Tired**	3 ± 1	3 ± 1	3 ± 1
**Sick to stomach**	1 ± 1	1 ± 1	1 ± 1
**Hot**	4 ± 1	4 ± 1	3 ± 1
**Trouble concentrating**	2 ± 1	2 ± 1	1 ± 1
**Chills**	1 ± 0	1 ± 0	1 ± 0

Superscript letters indicate statistical significance of post-ESQ symptoms from baseline to post-HA and from post-HAz to post-HA (*p* < 0.05).

**Table 2 ijerph-20-03219-t002:** Means and standard deviations (Ms ± SDs) of post-trial Environmental Symptoms Questionnaire (ESQ) total scores for maximal heat training group (HT_MAX_), minimum heat training group (HT_MIN_), and the control group (HT_CON_) after heat acclimation (post-HA), at week 4 of HT (post-HT4), and at week 8 of HT (post-HT8). Groups were split post-HA prior to beginning HT (post-HT4 and post-HT8). Group summaries were extrapolated from post-HA to help track environmental symptom changes that occurred over HT.

	Post-HA (M ± SD)	Post-HT4 (M ± SD)	Post-HT8 (M ± SD)
HT_MAX_	25 ± 4	26 ± 7	25 ± 3 ^#^
HT_MIN_	26 ± 4	32 ± 11	30 ± 7
HT_CON_	27 ± 7*	30 ± 13	30 ± 11 *^,#^

* indicates statistically significant differences (*p* < 0.05) within groups over time (post-HA and post-HT8). ^#^ indicates statistically significant differences (*p* < 0.05) between groups (HT_MAX_ and HT_CON_).

## Data Availability

The data from the current study are available in this manuscript. Additional data from the larger study are available at [22,23,24,25,26].

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
