# Peer review of "Environmental Stress Symptoms during Heat Acclimatization, Heat Acclimation, and Intermittent Heat Training"

_ijerph, 2023, doi:10.3390/ijerph20043219_

Round 1

Reviewer 1 Report

The aim of this paper was to examine changes in environmental symptoms during heat acclimatization, heat acclimation and intermittent heat training to determine whether there is a higher ESQ post trial vs pre trial and whether ESQ decreases incrementally post trial over the course of heat acclimatisation, heat acclimation and heat training. A secondary aim was to investigate the relationship between ESQ symptoms and thermal sensation, heart rate and rectal temperature to determine if a single or combined measure might be predictive of post-trial ESQ symptoms. This was a large study, with a large sample size conducted over a significant period of time. As with any heat adaptation/decay work, this required significant work and time investment by the researchers and they are to be commended for their efforts in completing this project and producing this work that adds valuable information to the field.

General points:

To me, the main issue is the difficult in following the reporting of the study and interpreting of the data. There is a natural order to the work in the sense that it is;

(a)    baseline to post heat acclimatization,

(b)    post heat acclimatization to post heat acclimation and also baseline to post heat acclimation

(c)     post heat acclimation to post heat training and also baseline to post heat training

However, I do not feel that the results are reported consistently in this order and therefore it is difficult to interpret important data in some areas. For example, in line 196 of the manuscript, there is reference to baseline to post HA being significantly different and post HAz to post HA but there is no reference to whether baseline to post Haz was significant when table 1 would suggest it is?  Perhaps a large table presenting the data for baseline, post heat acclimatization, post heat acclimation and post heat training would improve this element but also, a more consistent report in the prose would be beneficial.

A secondary point for consideration is clarity about the reporting of statistical analysis. In lines 290 – 294 the authors report the findings of Spearman’s correlational analysis. Usually, correlational analysis would report the ‘r’ value but in lines 290 – 294 the authors report r2. Can the authors please clarify the reasoning behind this and whether the values reported are r or r2.

A final point relates to terminology. Authors use the term ‘exercise trial’ to refer to what is essentially a heat stress test. Substituting ‘exercise trial’ with ‘heat stress test’ would be more consistent with much of the literature in this field and perhaps remove some of the difficulty with interpretation / comprehension.

Table 2:

If I understand correctly from the method section, the participants were split into HTmax, HTmin and HTcon groups after the Haz and HA. However, having table 2 with values for ESQ Post-HA suggests this splitting may have happened before. This needs clarification in the table as it is potentially misleading.

Specific points:

Line: 32:

Should this read ‘strain on athletes’ and not ‘stress on athletes’ as stress refers to any environmental factor that impacts the athlete whereas strain is the physiological response.

Line 146:

Can you provide any data to describe the HTmax, HTmin and HTcon groups? Were there any significant differences between these groups e.g. in body mass, body surface area, aerobic capacity?

Line 196 and line 351

By the absence of a reference to post ESQ from baseline to HAcz it suggests there was not a significant change in post ESQ due to training in a natural environment. Therefore, can the authors make the statement as per line 351?

Line 362

Change ‘twicer’ to ‘twice’

Line 370

Line 370 states that heat training 2x/week following Haz or HA is effective in improving and maintaining adaptations in environmental symptoms. The authors also state in line 331-332 that ESQ is not at this time a substitute marker for physiological adaptation but is a valuable addition…’ Given this and the fact (if I understand correctly) from table 2 that ESQ in Post-HT8 is not different to post-HT4 and no physiological data is presented for HA, HT4, HT8 by group, is this statement (a) appropriate / correct and (b) potentially misleading to practitioners. More evidence e.g. physiological is important before a decision can be finalised about the impact of HT for 8 weeks on the maintenance of adaptation.

Reviewer 2 Report

---General comments

I congratulate the authors for their effort in collecting the data. The strengths include good sample size, relevant physiological measurements (i.e. Trec), and longitudinal follow-up of athletes.

On the other hand, its weakness is considerable, preventing the understanding of the objectives of the study and its innovation. These include writing and the results presentation, but more importantly, the lack of relevant information.

The introduction is not clear, and the main gaps on the subject were not highlighted. The results are presented in a polluted way, and little care was taken in the preparation of tables and images. However, the point that stands out is the population used. I invite the authors to think about the use of runners and cyclists as the same sample. Although the characteristic "endurance" can fit into both aspects (i.e. long-term exercise, when applicable), I consider that body temperature can be influenced by the motor gesture and the amount of active muscles in these tasks. In this sense, I do not understand why they did not test the "exercise/modalide/sport" factor in the statistical analysis. I do not agree with the use of all athletes in the same sample. Other initial questions are presented below.

---Specific comments

--Introduction

-The introduction requires improvements and better streamlining. Although the title emphasizes the perspective of "Environmental Stress Symptoms", a large part of the introduction is intended to present the instruments used for this purpose. This layout is not elegant and takes the reader away from the main proposal. I recommend rewriting the introduction and focusing on the relationship between the highlighted symptoms and HAz/HA/HT, highlighting the scientific gaps in the literature on this scenario. Therefore, I recommend removing deep explanations about the scales, especially from your questions.

-Another point worth mentioning is the use of the term "athlete". Which athletes? Do the highlighted protocols (HAz / HA / HT) differ in terms of their benefits (Lines 71-76) when comparing different groups of athletes? Do the protocols present different results according to the place where they train? I recommend better targeting the population with which the study was developed in the introduction.

-Not least, I recommend synthesizing the sentences described in the introduction. Many are wordy and difficult to follow.

--Methods

-Lines 107-110 - Based on references 22 and 23, I understood that this is the third study arising from the same data collection. While this is not a problem, I urge caution for authors in the highlighted lines. Explore better which data were used in articles 22 and 23, avoiding future criticisms about plagiarism.

 -Line 112 – Remove “mean ± standard deviation [M±SD]” here and so on. Also, explain what sport or exercise refers to the “endurance” category.

-Line 114 – Is there some data to prove the “steady state exercise”? Which physiological variables were collected to prove it?

-Line 146 – Explain how these volumes were selected in the manuscript. Why more spaced volumes (i.e 5 vs 2 times per week) were not used?

-Line 161-162 – “Data are reported as (F-value, p-value).” ??

-Line 169-170 – Although this is not wrong, I recommend also using the p-value. If we consider thousands of pieces of information, a correlation of 0.3 can be relevant.

-Line 171-172 – “Data are reported as (r2-value, p-value).” ??

--Results

-Improve this entire section by removing the subtitles.

-Line 177-184 – This is the first time that the modalities or sports were presented. The detailed information regarding the collected variables during training must be presented in some well-elaborated figure. As mentioned previously, the use of two different sports must be considered by the authors.

-Improve the presentation of Tables. The explanation of the acronyms is missing in Table 2. Insert the p-value in the table.  

-Improve the presentation of the figures.

-Insert the p-value in the figure.

Round 2

Reviewer 2 Report

The authors made efforts to respond to my suggestions. The study was improved, but I highlight two relevant points in this process.

 1) Initially, I still recommend improving the quality of the tables and figures. Tables for example were probably pasted in image format. This drastically reduces its quality. The figures still have poor resolution. However, it is up to the authors to decide on this. Such a point will not be decisive for my evaluation.

 2) Unlike point 1, I strongly recommend inserting the absence of training control as a limitation. The authors responded that

"While this is really great point, and would be very interesting to analyze in itself, it is beyond the scope of this manuscript as the summer training period was not controlled for intentionally, and all participants utilized the same modality of treadmill running for the remainder of the study following HAz. The authors respect and appreciate this thought and point."

Honestly, I didn't understand the term "intentionally". Did you not follow the training on purpose? I see no reason for that. Also, whenever a certain intervention based on physical training takes place, load control is the basis for analyzing its efficiency. Without this information, replications with other populations are not feasible. Therefore, the authors' answer is not enough, and I recommend inserting the absence of this control as a considerable limitation of the study.
